# Proteomic, Metabolomic, and Lipidomic Analyses of Lung Tissue Exposed to Mustard Gas

**DOI:** 10.3390/metabo12090815

**Published:** 2022-08-30

**Authors:** Elizabeth Dhummakupt, Conor Jenkins, Gabrielle Rizzo, Allison Melka, Daniel Carmany, Amber Prugh, Jennifer Horsmon, Julie Renner, Daniel Angelini

**Affiliations:** 1US Army, Combat Capabilities Development Command (DEVCOM) Chemical Biological Center, BioSciences Division, Aberdeen Proving Ground, Edgewood, MD 21010, USA; 2Excet, Inc., Springfield, VA 22150, USA; 3US Army, Combat Capabilities Development Command (DEVCOM) Chemical Biological Center, Threat Agent Sciences Division, Aberdeen Proving Ground, Edgewood, MD 21010, USA

**Keywords:** sulfur mustard, lung, histology, metabolites, proteins, lipids, mass spectrometry, inflammation, DNA and RNA damage

## Abstract

Sulfur mustard (HD) poses a serious threat due to its relatively simple production process. Exposure to HD in the short-term causes an inflammatory response, while long-term exposure results in DNA and RNA damage. Respiratory tract tissue models were exposed to relatively low concentrations of HD and collected at 3 and 24 h post exposure. Histology, cytokine ELISAs, and mass spectrometric-based analyses were performed. Histology and ELISA data confirmed previously seen lung damage and inflammatory markers from HD exposure. The multi-omic mass spectrometry data showed variation in proteins and metabolites associated with increased inflammation, as well as DNA and RNA damage. HD exposure causes DNA and RNA damage that results in variation of proteins and metabolites that are associated with transcription, translation and cellular energy.

## 1. Introduction

Sulfur mustard (HD; mustard gas) or bis(2-chloroethyl) sulfide (C_4_H_8_Cl_2_S; CAS #: 505-60-2) is a vesicant that was developed as a chemical warfare agent (CWA) for battlefield use during World War I (1914–1918) [1]. It was initially deployed by Germany in 1917 and was later named the “King of Battle Gases” because it caused more chemical casualties than all other CWAs used during that war [2]. Since World War I, HD has been used in several different conflicts in both the 20th and 21st centuries, including the Japanese Occupation of China (1937–1945) [3], the Iran-Iraq War (1980–1988) [4], and most recently the occupation of Iraq by the terrorist group Islamic State of Iraq and the Levant (ISIL) (2014–2017) [5]. HD is currently considered a serious threat for use due to the large stockpiles still in existence and the relative ease of both acquiring the necessary materials and the production process [6,7].

HD exposure can be lethal at high concentrations, but, more often, cause incapacitating injuries that require extensive long-term medical care [8]. HD is a potent alkylating agent that irreversibly damages an individual’s DNA, and these changes can manifest themselves clinically even years after exposure [9]. In addition to DNA damage, HD irritates and blisters exposed areas, especially the eyes, skin, and respiratory tracts. These pathological changes typically take several hours to manifest (2–24 h) depending on the severity of exposure [8,10]. During a dermal exposure, much of the HD will be absorbed directly into the cells composing the skin; the absorbed amount can increase depending on environmental factors (e.g., humidity) or physiological factors of the exposed individual (e.g., extensive sweating) [8,10]. Once absorbed, HD induces localized inflammation, edema, and necrosis leading to the separation of the epidermal layer of skin from the dermis. This is observed clinically by formation of large blisters known as bullae [11,12]. Inhalation exposure, which is much more serious to the individual, induces similar pathological changes in the respiratory tract. Similar to the skin, exposure of airways also induces extensive inflammation, edema, and cellular necrosis resulting in the sloughing of the respiratory tract epithelium and pulmonary obstruction [9,13].

While many acute health effects from HD exposure are associated with DNA alkylation and the immediately resulting adducts, these DNA adducts would be recognized by cellular DNA damage response elements [14,15]. In the case of severe DNA damage, cells either initiate the DNA repair process or activate programmed cell death (i.e., apoptosis) [16]. While many cases of HD exposure recover completely, there are well-documented delayed and long-term health effects in some cases [17,18]. Recent studies propose that these chronic health effects are associated with epigenetic alterations [19,20].

To date, most of the studies on HD have been focused on genomic and transcriptomic analyses due to the DNA damage incurred upon exposure [16]. For this study, we will be utilizing proteomic, metabolomics and lipidomic analyses to elucidate further downstream markers of HD exposure. The combination of these three mass spectral-based techniques is known as “multi-omics”, which attempts to more holistically understand changes occurring across multiple molecule types. Multi-omic analysis integrates multiple data streams to allow a more comprehensive understanding of biological processes under different conditions [21]. To the best of our knowledge, this is the first study to use this approach to examine the pathological effects of HD.

For this study, we employed a three-dimensional respiratory tract tissue model derived from primary, normal human tracheal/bronchial epithelial cells cultured at the air-liquid interface (ALI) [22,23,24] for the HD exposures. This tissue model contains multiple cell types found within the respiratory tract including goblet cells, basal cells, Clara cells, and ciliated cells with actively beating cilia [22,24,25]; the cells in these tissues also contain functional tight junctions [26,27] as they do in vivo. It has recently been employed for predictive toxicology studies [23,24] as well as in the evaluation of new drugs for the treatment of lung diseases such as chronic obstructive pulmonary disease and asthma [28,29]. Overall, this model gives a more accurate and realistic representation of the human respiratory tract compared to traditional cell culture models.

## 2. Materials and Methods

### 2.1. In Vitro Lung Tissue Model

EpiAirway^TM^ three-dimensional human airway epithelium tissue models (AIR-100-HF; hydrocortisone-free; 9 mm insert; 0.4-µm pore size) were purchased from MatTek, Corp. (Ashland, MA, USA). These tissues consisted of fully differentiated human airway epithelium cultured at the ALI [22]. Upon receipt, the tissues were transferred to 6-well culture plates containing 1 mL fresh cell culture media (AIR-100-ASY; MatTek) and placed in the tissue culture incubator set at 37 °C/5% CO_2_ for 18–24 h prior to exposures.

### 2.2. Caution—Chemical Warfare Agent Notice and Handling

Due to the acute hazards with HD, all experiments involving HD were performed by qualified personnel in certified chemical fume hoods equipped with an advanced filtration system that protects the user and the environment at the Combat Capabilities Development Command (CCDC) Chemical Biological Center (CBC) according to all Federal, State, and International guidelines.

### 2.3. Sulfur Mustard (HD) Preparation and Exposure Procedures

The HD used in this study was synthesized and purified (99.6 ± 0.4 wt. %) by U.S. Army CCDC CBC chemists in accordance with international regulations. Neat HD was then diluted with corn oil (Sigma Aldrich; St. Louis, MO, USA) into 1 mL working stocks at concentrations of 8.3 mg/mL (52.2 mM), 9.16 mg/mL (57.6 mM), or 9.21 mg/mL (57.9 mM) and stored at 4 °C until use. EpiAirway^TM^ tissues were exposed on the apical side of the culture to increasing concentrations of HD diluted in corn oil (0.01–2.5 mg/mL; 0.0629–15.7 mM), vehicle (corn oil), 10% formalin (positive control; Fisher Scientific; Waltham, MA, USA), or air (negative control) for 3 h. At the conclusion of the exposure, the exposed materials were removed and the tissues were washed with 0.5 mL PBS (×3). For tissues analyzed at 3 h, the tissues were immediately processed for analysis as described below. For analysis at later time points (6–24 h), the tissues were placed back in the incubator and processed at the time points indicated. At the end of each time point, cell culture media was collected and stored at −80 °C for later analysis.

### 2.4. Determination of Cellular Viability

The MTT assay was used to determine cellular viability following exposure to HD (MatTek). Preparation and execution of this assay was performed according to the manufacturer’s stated protocol. The EpiAirway^TM^ inserts were quickly rinsed (top and bottom) with PBS and then placed into a 24-well MTT assay reagent plate and incubated at 37 °C/5% CO_2_ for 1.5 h. Just prior to end of this incubation period, 2 mL MTT extractant solution was dispensed in each well of a new 24-well plate. At end of the incubation period, the inserts were removed from the MTT assay reagent plate, blotted dry, and then transferred to the extractant plate. This plate was then wrapped with parafilm, covered with foil, and incubated overnight at room temperature. At the end of the overnight extraction period, the inserts were removed and discarded. The liquid contents of the extraction plate were then mixed and 200 µL of each well’s sample was added to a 96-well plate. For a background reading, 200 µL extractant solution was added to 3 wells of the 96-well plate. The plate was then read at 650 nm for background determination and at 570 nm for the assay reading on a SpectraMax plate reader. The results were displayed as a percentage of viability compared to untreated control.

### 2.5. Determination of Necrotic Cell Death

Analysis of the cell culture media following HD exposure was performed using the CyQUANT Lactate Dehydrogenase (LDH) Cytotoxicity Assay (Invitrogen; Waltham, MA, USA) to determine the presence of necrotic cell death in the exposed tissues [30]. The assay was performed according to the manufacturer’s specified protocol. Experimental samples were read on a SpectraMax Plate Reader and the results were expressed as a percentage of relative cytotoxicity compared to untreated control.

### 2.6. Histological Analysis

For histological analysis, the tissues were fixed in 10% formalin overnight at room temperature and then washed with PBS (×3). The fixed tissues were then paraffin-embedded using routine histological techniques. The paraffin-embedded tissues were cut into 6-µm sections and placed on clean charged glass slides. The slides were subjected to deparaffinization (100% xylene, 3 × 5 min at room temperature) and rehydrated by washes in decreasing concentrations of ethanol (2 × 100%, 1 × 95%, 1 × 70%, and 1 × 50% for 3–5 min each) followed by a single 5 min wash in tap H_2_O [31]. The rehydrated slides were stained with hematoxylin and eosin and then visualized using the EVOS M5000 Imaging System (ThermoFisher Scientific, Waltham, MA, USA).

### 2.7. Individual Cytokine ELISAs

Individual ELISAs for the cytokines IL-1α, IL-1β, IL-6, and TNFα were purchased from R&D Systems (Minneapolis, MN, USA) and performed according to the manufacturer’s recommended protocols. Briefly, cell culture media from untreated, vehicle (corn oil), and HD (0.01 mg/mL, 0.1 mg/mL) were examined in triplicate from both 3 and 24 h time points for each of the cytokines stated above. A final reading of the microtiter plate was conducted at 450 nm using a SpectraMax plate reader. Results were expressed as mean ± standard error of the mean (SEM) in pg/mL for each of the individual cytokines examined.

### 2.8. Cell Viability/Cytotoxicity Statistical Analysis

Cell Viability/cytotoxicity data was analyzed using GraphPad Prism 8 software (GraphPad Software; San Diego, CA, USA). The experimental values in this study were expressed as means ± SEM. Analysis of variance (ANOVA) was used to compare the mean responses between experimental and control groups for experiments containing multiple groups. The Tukey multiple comparison test was used to determine if statistically significant differences existed between groups. For these studies, a *p* value of <0.05 was considered statistically significant.

### 2.9. Preparation of EpiAirway Tissues for Multiomic Analysis

At the specified time point, the tissues were removed from the inserts using an 8 mm biopsy punch, placed in centrifuge tubes (Eppendorf Lo-Bind, 2 mL centrifuge tubes), and then flash frozen in liquid N_2_. The tubes were then stored at −80 °C until further processing. Then, 1 mL of ice-cold Optima™ methanol (Fisher Scientific) with 0.1% Optima™ formic acid (Fisher Scientific) was added to each tube containing a tissue sample. Keeping the tubes on ice, each sample was ground using a Potter tissue homogenizer for 1–2 min. The samples were then centrifuged for 10 min at 20,000× *g* at 4 °C. The supernatant was then removed to a glass vial for lipid/metabolite extraction, and the protein pellet was stored at −80 °C for protein extraction.

### 2.10. Lipid/Metabolite Extraction

One mL of methyl-tert-butylether (MTBE, Fisher Scientific) was added to the supernatant in each glass vial and mixed via tabletop vortex for 30 s. Subsequently, 1 mL of mass spectral grade water (Fisher Scientific) was added, and the samples mixed again for 30 s. Samples were then allowed to incubate at 4 °C for 1 h. After incubation, the samples were centrifuged at 1000× *g* for 5 min. After centrifugation, the aqueous (lower; metabolite) layer was removed into Lo-Bind Eppendorf tubes and were dried overnight in a speed-vac. The organic (upper; lipid) layer was transferred to glass LC vials and 2 µL of Avanti Splash II Lipidomix (Birmingham, AL, USA) were added for an internal standard. The glass vials were stored at −80 °C until mass spectral analysis.

### 2.11. Dansylation Assay and Metabolomics Normalization

The dansylation protocol has been previously published in organ-on-a-chip samples [32]. Briefly, the metabolites were resuspended in 50 µL of Fisher Scientific Optima™ water with 0.1% formic acid. Then, 15 µL of metabolomics sample were removed to a second Lo-Bind Eppendorf tube and dried down in a speed-vac and then resuspended in 75 µL Optima™ water. An additional 75 µL 0.5 M sodium bicarbonate was added to each sample, followed by addition of 75 µL of an 18 mg/mL dansyl chloride in acetonitrile solution, made fresh daily, kept on ice, and protected from light. Samples were then incubated for 1 h in a shaking incubator at 60 °C (350 rpm). The labeling reaction was quenched by the addition of 15 µL 250 mM sodium hydroxide, and incubated an additional 10 min at 60 °C (350 rpm). Then, 75 µL 1.6% formic acid in 50% acetonitrile was added to each sample. Next, 210 µL of each labeled sample was removed and placed into a new tube, and 630 µL ethyl acetate was added to extract labeled metabolites. Samples were then vortexed for 30 s, then spun down in a tabletop micro centrifuge for 2 min at 5250× *g*. Then, 200 µL was removed from the upper ethyl acetate fraction and deposited into the well of a PerkinElmer (Waltham, MA, USA) Spectra Plate 96-well plate. Each standard was read in duplicate at a wavelength of 340 nm.

Metabolomics samples were normalized to the sample with the lowest concentration based on the dansylation assay. All samples were normalized by diluting to the lowest-concentrated sample in Optima™ water with 0.1% Optima™ formic acid. For these samples, 2 µL of internal standard was added to each sample prior to mass spectral analysis. The internal standard is prepared by making working stocks of each solution at 2 mg/mL by dissolving 10 mg of each standard in 5 mL of 90:10 water: acetonitrile. A working stock solution was prepared by combining the following volumes of each internal standard stock into a single vial containing 4715 µL of Fisher Optima gold label water with 0.1% formic acid (final volume 5000 µL): d3-creatine (10 µL), d10-leucine (10 µL), d3-L-tryptophan (10 µL), ^13^C6-citric acid (20 µL), ^13^C11-tryptophan (100 µL), ^13^C6-leucine (10 µL), ^13^C6-L-phenylalanine (10 µL), T-BOC-L-tert-leucine (10 µL), and T-BOC-L-aspartic acid (5 µL).

### 2.12. Proteomics Sample Preparation and Normalization

The protein pellets were each resuspended in 500 µL lysis buffer (4% SDS + 50 mM ammonium bicarbonate), vortexed 20 s, and incubated at 95 °C for 5 min. Protein concentration was determined using a Pierce BCA Protein Assay Kit (ThermoFisher Scientific). Protifi (Farmingdale, NY, USA) S-Trap Mini columns were used to digest samples, according to manufacturer’s protocol. Briefly, samples were reduced by addition of 20 mM DTT at 95 °C for 10 min, followed by alkylation via addition of 0.1 M iodoacetamide to a final concentration of 0.1 M for 30 min, followed by acidified via addition of 1.5% formic acid. Finally, samples were mixed with 6x volume S-Trap buffer (90% methanol with 100 mM TEAB) and loaded into S-Trap Mini columns. Trypsin/Lys-C (Promega, Madison, WI, USA) was added at a 1:20 trypsin: protein ratio based on results of the BCA assay. Digestion occurred overnight at 37 °C in a stationary incubator. Pierce (ThermoFisher Scientific) Quantitative Colorimetric Assay was used to confirm peptide concentration. *Detailed mass spectral methods are available in the Appendix A [33,34,35,36,37]*.

## 3. Results

### 3.1. Histology and Cellular Viability

We examined the pathological effects of HD on a cultured human respiratory tract tissue using a multi-omic approach (e.g., proteomics, lipidomics, metabolomics). Following a series of dose- and time-dependent HD exposures, we initially evaluated the viability as well as the associated cytotoxicity and histopathologic changes of the cells in this model to determine the appropriate dose and time requirements for the multi-omic analyses. The viability studies revealed a dose-dependent reduction in cellular viability 24 h post-HD exposure (0.05–2.5 mg/mL) with a concomitant increase in cytotoxicity as measured by an LDH release assay. In addition, histological analysis (0.1 mg/mL) revealed sloughing of the apical layer of cells similar to what is observed following in a clinical respiratory exposure [8,33,34].

Figure 1 shows hematoxylin and eosin (H&E) staining of treated and untreated cultured lung tissue. The left image is of control tissue with all epithelial layers intact. The right image has been exposed to HD and much of the ciliated surface has sloughed off and the mucociliary epithelium has collapsed. In addition, several of the remaining cells display what appears to be both (1) condensed or pyknotic nuclei indicating the occurrence of apoptosis and (2) fragmented nuclei or karyorrhexis, which is a later state of apoptosis.

Figure 2 shows the percentage of cellular viability from exposure to increasing concentrations of HD at three hours post-exposure (Figure 2A) and 24 h post-exposure (Figure 2B). While there is no significant reduction in cellular viability at three hours post-exposure for any of the tested concentrations, there is significant reduction in cellular viability for all exposures of 0.05 mg/mL concentration and greater at the 24 h post-exposure time point. These results corroborate previously observed data that evidence of HD exposure does not physically manifest until 24 h post-exposure. Complimentary cytotoxicity data can be seen in Appendix A.

### 3.2. Proteomic Data

Proteomics results were filtered by high FDR confidence protein identifications, which resulted in 4853 high confidence (≤0.01 FDR, ≥1 Unique Peptide, >1 PSMs) proteins being identified. At 3 h post exposure, 724 proteins were down-regulated and 655 were up-regulated significantly (Figure 3A). At 24 h post exposure, 2045 proteins were down-regulated and 527 were up-regulated significantly (Figure 3B). Principal component analysis (PCA) was performed on the control samples and HD-exposed samples at both the 3- and 24 h post-exposure time points. As seen in Figure 3C, there is separation between the 3 h post-exposure control and exposed samples, indicating molecular changes are occurring at that early time point. When the 24 h control and exposed samples are added, the greatest separation is between the 24 h exposed samples and all the other samples, as seen in Figure 3D. Performing a chromosomal position mapping of the significantly downregulated proteins at 3- and 24 h post exposure show a chromosome wide effect of the HD on the genome in Figure 4. The severity of the downregulation is increased at the 24 h mark.

Exposure to HD elicits an inflammatory response, which is reflected in the dysregulation of certain systems. One such system is the plasminogen activator system, whose proteins have been utilized as a non-specific biomarker for inflammation [35]. Figure 5A depicts plasminogen activator inhibitor 2 (PAI2), and it is significantly downregulated at 3 h post-exposure and significantly upregulated at 24 h At the 3 h timepoint, urokinase-type plasminogen activator receptor (uPAR) is slightly down regulated (Figure 5B) due to the negative feedback regulation by urokinase-type plasminogen activator (uPA) cleavage of uPAR during plasminogen production showing the initiation of the cellular inflammatory response. This changes at the 24 h timepoint where there is a significant upregulation of uPAR over baseline uPA shows a similar trend where the initial inflammatory response is associated with a slight decrease in uPA at the 3 h timepoint (Figure 5C) where it is cleaved during zymogen plasminogen activation.

uPA also acts as a serine protease that activates matrix metalloproteinase-9 (MMP-9), and, therefore, indicates an inflammatory response [36]. MMP-9 is a marker of inflammation, tissue remodeling, wound healing, and mobilization of tissue-bound growth factors and cytokines [37,38,39]. A previous study showed that changes in the dynamic equilibrium between MMP-9 and TIMP1 can be indicative of acute lung injury [40]. Figure 6A and Figure 6B shows significant dysregulation versus the control groups. Our findings are complementary to these previous findings as the ratio between MMP-9 and TIMP1 first increases at the 3 h time point followed by subsequent decrease at the 24 h time point showing damage of the extracellular matrix of the lung cells.

Other proteins associated with inflammation were also observed (Appendix A). Tissue factor pathway inhibitor (TFPI) is an anticoagulant protein and thought to be an anti-inflammatory protein due to its ability to limit thrombin generation [41]. The dysregulation of TFPI at 24 h indicates an increase in inflammation within the system.

Pathway network mapping was performed, and Figure 7 depicts the pathways that were enriched in the 24 h post-exposure samples. Figure 7A are the pathways that are up-regulated, and Figure 7B are the pathways that are down-regulated. The closely related pathways are shown grouped together with interconnecting lines. Several of the significantly changing upregulated proteins identified were mapped to the base excision repair (BER) pathway, which is seen in Figure 8. Figure 8A shows the BER pathway at 3 h post-exposure, and a few of the proteins are up-regulated compared to baseline (seen in yellow, orange, and red colors). These proteins are primarily polymerases and function as part of DNA repair (POLB, POLL) and DNA synthesis during DNA repair (POLE). Figure 8B shows the BER pathway at 24 h post-exposure, and several proteins, including those at the 3 h time point, are upregulated. Additional proteins that are upregulated are poly(ADP-ribose)polymerase (PARP), proliferating cell nuclear antigen (PCNA) and DNA-(apurinc or apyrimidinic site) endonuclease 2 (APEX2). At both time points, the high mobility group protein B1 (HMGB1) is upregulated, and this protein promotes host inflammatory response and is also involved in coordination and integration of innate and adaptive immune responses [42].

Another pathway with significantly upregulated proteins is the mRNA surveillance pathway, which is in Figure 9. This pathway uses a variety of mechanisms to detect abnormalities in mRNAs and a variety of RNA-degradation enzymes [43]. Several upregulated proteins are associated with processing the pre-mRNA into mRNA, such as those that add the 5′ cap and the poly-A tail to the 3′-end. Additionally, within the nucleus, the TDP-43 protein was upregulated, which also processes mRNA, specifically in alternative splicing [44].

### 3.3. Metabolomic and Lipidomic Data

At 3 h post-exposure, approximately 22 metabolomic features were found to be significant with reverse phase (RP) separation, but at 24 h, approximately 210 metabolomic features were found to be significant. With HILIC separation, approximately 40 metabolomic features were found to be significant at 3 h post-exposure and 132 metabolomic features were found to be significant at 24 h.

Upon analyzing the 24 h post-exposure data significantly changing metabolites, several were identified and mapped to the nucleotide metabolism pathway, particularly those focused around adenosine metabolism. This can be seen in Figure 10, in which several metabolites are seen to be down-regulated at 24 h post-exposure. Adenosine levels can fluctuate based upon stress [45] and can act as feedback regulators of inflammation [46]. Table 1 shows most of the features in the adenosine metabolism group, most of which are significantly down-regulated and significant.

Lipidomics was performed which several lipids identified as significantly changing between exposed and control samples. Specifically, sphingomyelins (SM) and ceramides (Cer) were identified, which are subclasses of sphigolipids. These two subclasses are depicted as box-and-whisker plots in Figure 11. Figure 11A compares the abundance of sphingomyelins of 3 h exposed, 3 h control, 24 h exposed and 24 h control samples while Figure 11B compares the abundance of ceramides in these samples. Based on these box-and-whisker plots, sphingomyelins are down-regulated at both the 3- and 24 h post-exposure time points, and the ceramides are down-regulated 3 h post-exposure and back to “normal” levels 24 h post-exposure. As part of the breakdown of sphingomyelin by sphingomyelinase, ceramide is generated when the phosphocholine head group is removed [47].

## 4. Discussion

### 4.1. Inflammation

Exposure to HD can directly affect lung morphology. When inhaled in sufficient amounts, HD induces apoptosis and/or necrosis of the respiratory tract epithelium resulting in the sloughing of this lining [6,7,8]. This process can lead to specific negative clinical outcomes, such as pulmonary obstruction and increased susceptibility to secondary infection [7]. This pathological process has been observed in both clinical and experimental models of HD inhalation injury [48]. In Figure 1, EpiAirway^TM^ tissue exposed to HD displayed a loss of several layers of epithelial cells including the ciliated surface. In addition, several pyknotic and karyorrhexes nuclei were observed in the HD-treated tissues, indicating the presence of apoptotic cells. It is interesting to note that several of the HD-exposed tissues displayed complete separation of the cells from the membrane which mimics the cellular sloughing that occurs following clinical observation to exposures (Appendix A). Overall, these results indicate that the in vitro EpiAirway^TM^ model mimics the histopathological alterations associated with an in vivo exposure to HD.

HD exposure causes inflammation injuries within lung tissue, and significant changes in several proteins aligned with inflammatory response. As depicted in Figure 5A, the PAI2 protein is downregulated at 3 h post-exposure. PAI2 is known to be induced by proinflammatory mediators and protects cells from TNF apoptosis [49]. At 3 h post-exposure, the inhibition of the binding of uPA and its receptor uPAR by PAI2 is suppressed to encourage tissue healing from the damage of HD exposure. However, at 24 h post-exposure, this changes significantly as upregulation of proteins involved in TNF-α-mediated apoptosis (TNFAIP3, TNFSF9, TNFRSF10A, and TNFRSF10B) induces expression of PAI2 to prevent the cellular death. Additionally, the uPAR protein shown in Figure 5B becomes significantly upregulated at 24 h versus the control baseline, and this upregulation is consistent with previous research on states of prolonged inflammation that occur in chronic obstructive pulmonary disorder (COPD) and other bacterial, viral, and chemical exposure events [50,51,52,53].

MMP-9 is part of a family of zinc-dependent endopeptidases that are responsible for multiple roles with human physiology. However, for the scope of this work, the upregulation of MMP-9 reflects the inflammation and immune response occurring within the lung system [54]. MMP-9 activates additional inflammatory cytokines, such as tumor necrosis factor α (TNF-α), interleukin 1β (IL-1β), and transforming growth factor β (TGF-β). It is interesting to note that we had observed an increase in the inflammatory cytokines IL-1α, IL-1β, IL-6, and TNF-α in the tissue culture media following HD exposure (Appendix A). We also observed the dynamic equilibrium between MMP-9 and TIMP1, which has also been previously indited for acute lung injury [36]. It is possible that the upregulation of MMP-9 in this model may be one of the driving factors causing an increase in the expression levels of these inflammatory cytokines.

Performing a GO enrichment analysis on the results (ShinyGO 0.76, Appendix A) indicate an enrichment of upregulated proteins at the 3 h time point associated with intraciliary transport involved in cilium assembly. The cilium is important in extracellular signal interpretation, regulating growth and development [55]. They also work in concert with airway mucus for mucociliary clearance [56]. After the tissue is exposed to HD, this clearance mechanism should be activated to remove the chemical from the lungs, leading to coughing of an individual exposed to the agent. The protein classes enriched for by the significantly upregulated proteins at 24 h show a family of organization pathways linked to collagen fibril organization (Figure 7). Healthy lung tissues maintain a fine balance between collagen production and degradation [57]. This leads to a state of pulmonary fibrosis that has also been observed post-HD exposure [58] where the lungs are going through a state of aberrant remodeling in response to inflammatory conditions.

Phosphocholine is an intermediate during the production of phosphatidylcholine whose low levels characterizes an enhanced inflammation state. This intermediate was observed to be significantly depleted in the 24 h post exposure samples (Log_2_FC −2.04, *p*. value 1.5 × 10^−4^, adj. *p*. value 7.8 × 10^−4^). At the 24 h time point, acetylcholine (Log_2_FC −2.50, *p*. value 1.7 × 10^−5^, adj. *p*. value 1.3 × 10^−4^) and acetylcarnitine (Log_2_FC −2.84, *p*. value 2.7 × 10^−6^, adj. *p*. value 3.9 × 10^−5^) are also decreased (Appendix A) which are indicative biomarkers for inflammation [59,60,61].

### 4.2. Proteins and Metabolites Related to DNA/RNA Damage

While proteins commonly associated with inflammation are affected due to HD exposure, additional proteins that specialize in DNA repair pathways are also significantly altered. HD is a known DNA alkylating agent and forms both monoadducts and inter-strand cross links [62,63]. Unrepaired monoadducts could react with nucleophilic proteins or with bases on the same or compliment DNA strand to form intra- or inter-strand cross links [15].

These HD-induced lesions would be expected to affect the DNA directed processes of replication and transcription, and the results from this study and others have confirmed this expectation [64]. Work by Masta et al. demonstrated a relationship between increasing concentrations of HD with decreasing transcriptional elongation in exposed *E.coli* cultures. Their work observed a complete cessation of transcription at 400 µM HD. The work detailed here utilized HD concentrations at approximately 600 µM or 0.1 mg/mL, and over 50% of the cells were still viable 24 h after exposure, as seen in Figure 2. Therefore, it can be surmised, that this concentration of HD in humans does not completely block transcription, but the protein and metabolite data indicate that these pathways are significantly affected.

Of the upregulated proteins at 24 h, ribonucleotide, ribose phosphate, and ribonucleotide biosynthetic and metabolic pathways were highly enriched for (Appendix A). These data provide additional evidence of the DNA and RNA damage that arises from HD exposure leading to an increase in production of these compounds to counteract the effects of HD.

The down regulated pathways identified provide additional data after HD exposure. At the 3 h point, there is no significant enrichment in any GO pathway. This also reflects the delayed symptomatic onset after HD exposure that has been observed. Conversely, at the 24 h time point on the other hand, transcription, translation, and splicing pathways are enriched in an interlinking family by network analysis, as seen in Figure 7. In addition, a subfamily associated with translation and transcription were also enriched, demonstrating a potential effect of the HD modified nucleobases to disrupt polymerase during elongation and termination phases as well as inhibiting the tRNA binding at the A site of the ribosome during translation. GO enrichment analysis of the significantly changing metabolites also showed the tRNA metabolic process pathway was downregulated (data not shown).

As demonstrated in Figure 8 and Figure 9, proteins within biochemical pathways associated with DNA repair and mRNA quality control are significantly upregulated 24 h after HD exposure. Previous research has hypothesized specific residues are preferentially alkylated by HD, specifically adenine and guanine. Masta et al. observed transcriptional blockages at sites of 5′-GG, 5′-AA, and 5′-GNC sequences on the DNA template strand. It was also observed that HD has a higher tendency to alkylate adenine residues.

Many proteins associated with mRNA degradation, like Dom34, PABP1, and MSL1 are up-regulated in the mRNA surveillance pathway. These proteins are involved in translation stalling [65], the recognition of premature termination codons, and mechanisms that protect cells from DNA damage-induced apoptosis and promotes cell survival from DNA-damaging agents [66]. These proteins and others point to the difficulty of translating the alkylated DNA into functional mRNAs for further protein synthesis. At these instances, the mRNA undergoes no-go decay (NGD), which degrades mRNAs containing stalled ribosomes [67]. NGD is caused when translational elongation is blocked in the presence of stable RNA structures [68], enzymatic cleavage [69], or chemically damaged sequences [70], which happens upon exposure to HD.

The preference for HD to alkylate specific residues is observed in the metabolomic data, and the significant downregulation aspects of nucleotide metabolism pathways, specifically those related to adenine. Nicotinamide is a precursor to NAD+ which has been shown to have a central role in HD-induced cytotoxicity. through the PARP1 pathway associated with elevated DNA damage and the NF-kappa Beta’s response to inflammation. Here, we observe a distinct downregulation of nicotinamide (log_2_FC −2.00, *p*. value 2.1 × 10^−5^ adj *p*. value 1.6 × 10^−4^) at the 24 h post-exposure time point (Appendix A) indicating a potential increase in NAD+ utilization in response to the exposure [71].

## 5. Conclusions

In this study, we demonstrate the utility of three-dimensional respiratory systems to model exposure of HD and the ability to utilize both histological staining and multi-omic biomarker analysis for biological response. Inflammatory markers from both cytokine panels and proteomic expression profiles observed replicate previous studies. Additional proteomic and metabolomic data indicate considerable DNA and RNA damage and the extensive depletion of adenine caused by HD exposure. Future analysis will look determine if any lipid changes occur from HD exposure.

## Figures and Tables

**Figure 1 metabolites-12-00815-f001:**
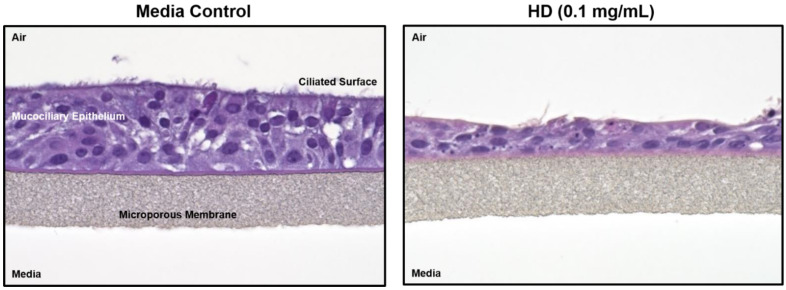
Histology changes associated with HD exposure in EpiAirway^TM^ tissues. Paraffin-embedded sections from EpiAirway^TM^ tissues exposed to media or HD (0.1 mg/mL; 0.629 mM) were rehydrated and stained with hematoxylin and eosin.

**Figure 2 metabolites-12-00815-f002:**
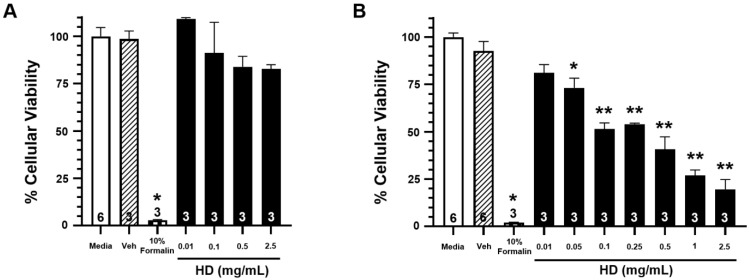
Effects of HD on the viability of EpiAirway^TM^ tissues. Open bars represent mean ± SEM of cellular viability of untreated EpiAirway^TM^ tissues at 3 h (**A**) or 24 h (**B**).Closed bars represent mean ± SEM of cellular viability of EpiAirway^TM^ tissues at 3 h (**A**) or 24 h (**B**) following treatment with increasing concentrations of HD (0.01–2.5 mg/mL; 0.0629–15.7 mM). Cross-hatched bars represent mean ± SEM of cellular viability of EpiAirway^TM^ tissues at 3 h (**A**) or 24 h (**B**) with an equivalent amount of vehicle (corn oil). Checkered bars represent mean ± SEM of cellular viability of EpiAirway^TM^ tissues treated with 10% formalin (positive control) at 3 h (**A**) or 24 h (**B**). Experimental n is indicated within each bar. * Significantly decreased compared to vehicle control at *p* < 0.05. ** Significantly decreased compared to vehicle control at *p* < 0.01. SEM indicates the standard error of the mean.

**Figure 3 metabolites-12-00815-f003:**
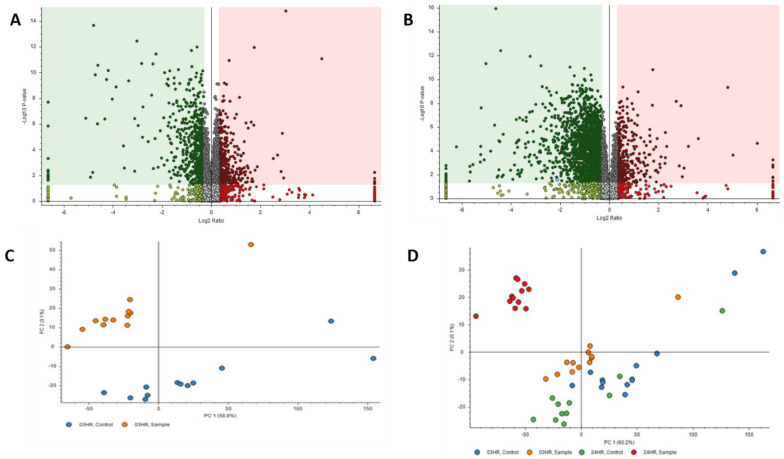
Volcano Plots of proteins at (**A**) 3 h post exposure and (**B**) 24 h post exposure. Principal component analysis (PCA) plots for (**C**) exposed and control samples at 3 h post exposure and (**D**) exposed and control samples at 24 h post exposure.

**Figure 4 metabolites-12-00815-f004:**
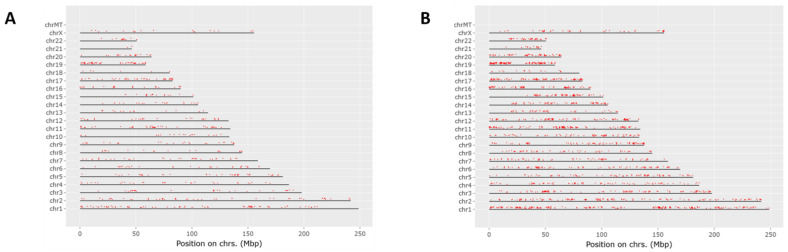
Chromosomal mapping of down regulated proteins at the 3 h mark (**A**) and the 24 h mark (**B**). This shows a wide chromosomal effect of HD exposure over time leading to the dysregulation of multiple proteins.

**Figure 5 metabolites-12-00815-f005:**
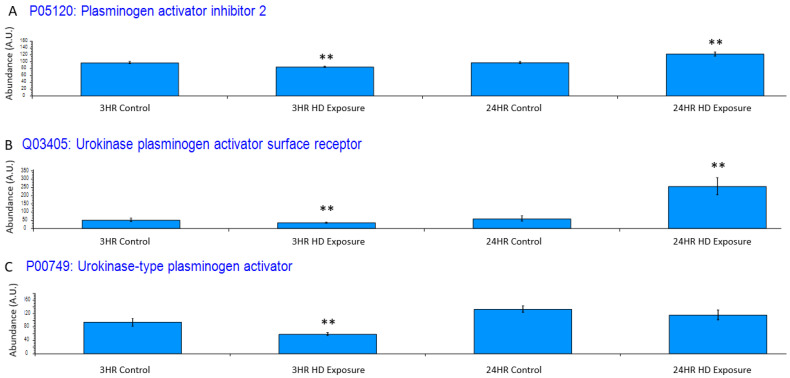
Bar graphs of the grouped abundances of (**A**) plasminogen activator inhibitor 2 (PAI2) expression profile across the control groups and the 3 h and 24 h exposures, (**B**) Urokinase plasminogen activator surface receptor dysregulation at the 3 h and 24 h marks indicating an upregulation in recruitment of urokinase-type plasminogen activator (PLAU) at the 24 h timepoint and initial inflammatory response at associated with the decrease in uPAR. The dysregulation appears to increase post exposure to a point of significant dysregulation, (**C**) PLAU shows increase expression indicating an inflammatory response. Significant dysregulation (adj. *p*-value ≤ 0.05) is shown by **.

**Figure 6 metabolites-12-00815-f006:**
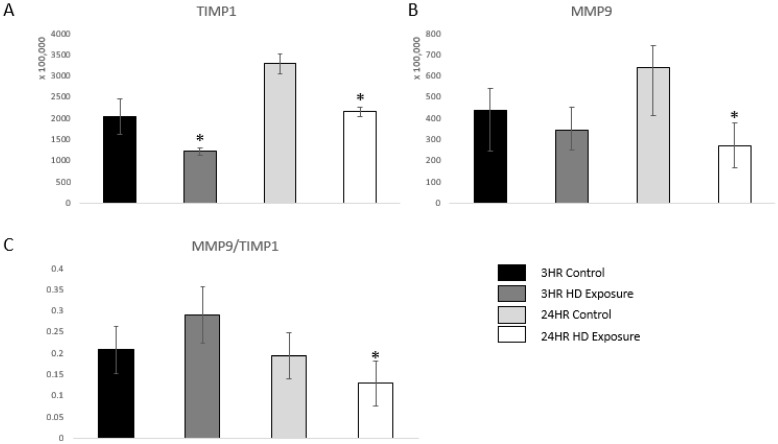
Bar graphs of the normalized abundances of (**A**) TIMP1 expression profile across the control groups, 3 h and 24 h exposures, (**B**) MMP-9 at the 3 h and 24 h marks, and (**C**) Ratios between the expression levels of MMP-9 vs. TIMP1. Significant dysregulation of TIMP1 and MMP9 (adj. *p*-value ≤ 0.05) is shown by * and significant changes in the ratio (*p*-value ≤ 0.05, students *t*-test) are indicated by *.

**Figure 7 metabolites-12-00815-f007:**
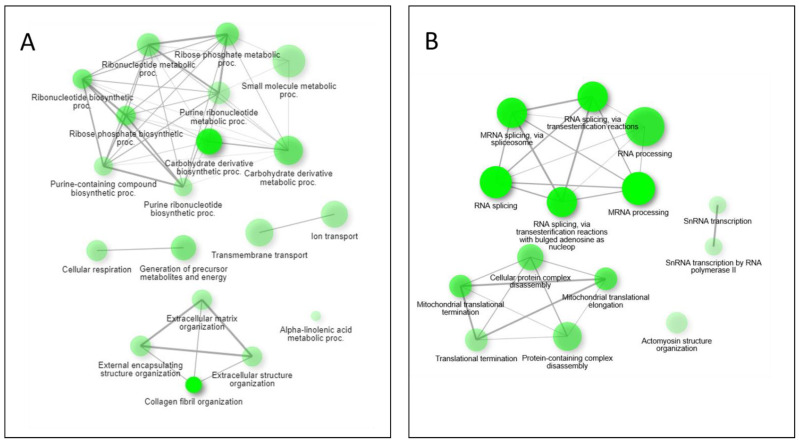
Pathway network mapping of enriched pathways upregulated (**A**) and downregulated (**B**) at the 24 h mark. Darker nodes are more significantly enriched gene sets. Bigger nodes represent larger gene sets. Thicker edges represent more overlapped genes.

**Figure 8 metabolites-12-00815-f008:**
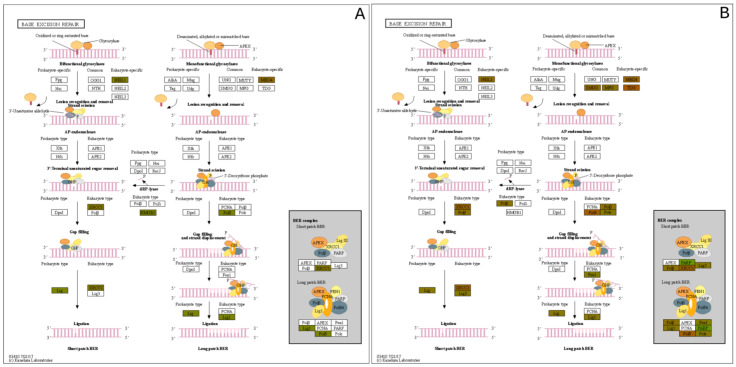
Base Excision Repair (BER) Pathway expression profiles of the (**A**) 3 h exposure and the (**B**) 24 h post exposure.

**Figure 9 metabolites-12-00815-f009:**
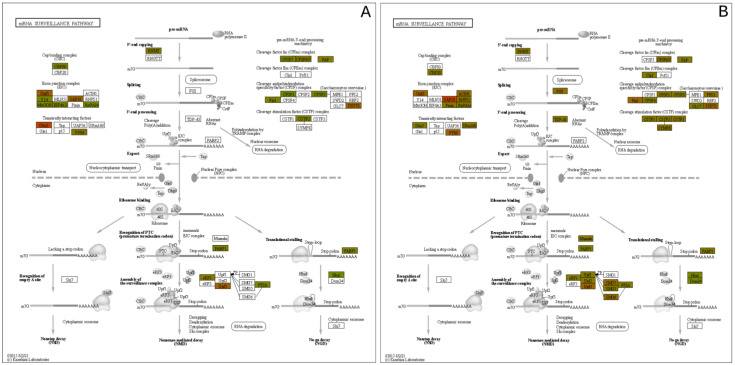
mRNA Surveillance pathway expression mapping at the (**A**) 3 h and (**B**) 24 h post exposure levels.

**Figure 10 metabolites-12-00815-f010:**
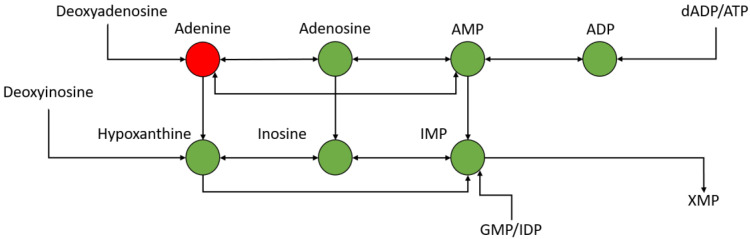
Portion of nucleotide metabolism pathway around adenine metabolism—green is down-regulated and red is up-regulated.

**Figure 11 metabolites-12-00815-f011:**
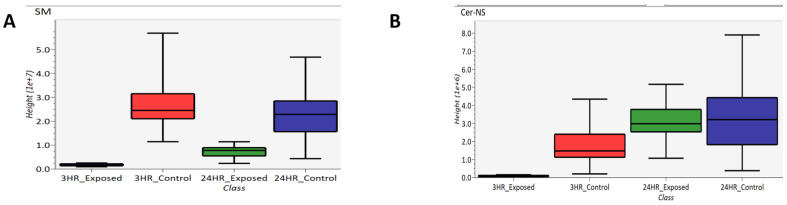
Box-and-whisker plots of (**A**) sphingomyelin and (**B**) ceramide.

**Table 1 metabolites-12-00815-t001:** Fold changes and *p*-values of significantly changing features in the nucleotide metabolism pathway 24 h post exposure.

Feature	Log_2_ Fold Change	*p*-Value	Adjusted *p*-Value
Adenine	0.31	0.05	0.09
Adenosine	−1.33	0.0099	0.0032
AMP	−1.41	0.00290	0.0078
ADP	−2.51	0.00041	0.0016
IMP	−5.94	2.1804 × 10^−8^	1.6326 × 10^−6^
Inosine	−3.42	3.3710 × 10^−8^	2.2415 × 10^−6^
Hypoxanthine	−2.69	1.2348 × 10^−5^	0.0001096

## Data Availability

The mass spectrometry proteomics data have been deposited in the ProteomeXchange Consortium, the reader can contact corresponding authors for raw data, due to the accession number from the PRIDE database has not been obtained. The mass spectrometry metabolomics data have been deposited to the MetaboLights [72] database (http://ebi.ac.uk/metabolights/) with the identifier MTBLS5467.

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
