# Peer review of "Proteomic, Metabolomic, and Lipidomic Analyses of Lung Tissue Exposed to Mustard Gas"

_metabolites, 2022, doi:10.3390/metabo12090815_

Round 1

Reviewer 1 Report

I have carefully read the work entitled "PROTEOMIC, METABOLOMIC, and LIPIDOMIC ANALYSES OF LUNG TISSUE EXPOSED TO MUSTARD GAS". The MS is clearly written and many details are given. I found it interesting, just a little too long.

Comments:

The last paragraph of the introduction should be moved to materials and methods because it explains and gives details of the samples used. 

The "materials and methods" section is too long. The details of the mass techniques should be shortened and move the details in the supplementary materials. 

Finally, many metabolic maps are shown but I did not see any mass spectra reported and commented on. Not even in the supplementary material. I need to take it for granted that everything that you found is quoted and it is extracted from the multi-omics mass analysis? 

I would like to see spectra with details on it. For each part: proteomic, metabolomics and lipidomics. You can add the spectra in the supplementary part. 

Author Response

The authors thank the reviewer for their comments. Specific answers to their point-by-point are below.

The last paragraph of the introduction should be moved to materials and methods because it explains and gives details of the samples used. 

The authors agree with this and have moved the last intro paragraph to the first paragraph of the materials and methods.

The "materials and methods" section is too long. The details of the mass techniques should be shortened and move the details in the supplementary materials. 

The mass spectrometry methods have been moved to supplementary materials.

Finally, many metabolic maps are shown but I did not see any mass spectra reported and commented on. Not even in the supplementary material. I need to take it for granted that everything that you found is quoted and it is extracted from the multi-omics mass analysis? 

In most multiomic mass spectrometry-based literature, images of mass spectra are not typically included as no real information can be gleaned from just looking at a spectral image alone. Untargeted metabolomic spectra consist of thousands of peaks that, even to an expert’s eye, look like grass. If this manuscript were detailing specific breakdown metabolites of sulfur mustard, it would be more appropriate to include a mass spectrum that depicts the peaks associated with the breakdown products.

For this untargeted multiomic analysis, the hundreds of data files are analyzed according to the data analysis methods, software and parameters detailed in the supplementary material and methods. Additionally, as part of this submission (and at other journals), spectra files are placed in publicly accessible databases for others to reproduce results based off parameters presented.

I would like to see spectra with details on it. For each part: proteomic, metabolomics and lipidomics. You can add the spectra in the supplementary part. 

Please refer to the previous reply. There would be no information added to the manuscript by including these spectra. The data will be made available via various databases, and all the spectra will be in those files.

Reviewer 2 Report

Thorough evaluation of cellular and molecular respiratory toxicology of a very well known chemical warfare agent at low dosage using a relatively new multimodal method. Perhaps a bit long and possibly some methods  could be put in appendices. I note the special issue is on cardiovascular effects while this study is on respiratory toxicology.

Author Response

The authors thank the reviewer for their comments. We would like to note that the mass spectrometry methods have been moved to the supplemental.

Reviewer 3 Report

In the present paper Dhummakupt and colleagues focused on the adverse effect of sulfur mustard (HD) exposure in a respiratory tract tissue model. The authors provided a multiomics approach to dissect the main changes due to 3h and 24h post-exposure. Their data allowed to identify in inflammation and DNA and RNA damage the primary driver of the damage induced by HD.

Although the topic is of interest, some issues need to be addressed before considering the paper suitable for publication.

MAJOR ISSUES

1.       Lines109-110: How did the authors chose the HD concentration values used for their analysis? Do they recapitulate a mean or acute exposure?

2.       Lines 340-342: It would be suitable if authors could corroborate the indication of the apoptosis by means of appropriate assays (e.g., TUNEL)

3.       Lines 394-405: Why did the authors consider only dysregulation of TFPI, MMP-9 and PLAU among those altered by HD exposure resulting from proteome analysis? Please, motivate the rationale and add it into the manuscript.

4.       Lines 394-405: Given the increase in MMP-9 protein levels, it would be of interest if the authors could provide indications about the expression levels of TIMP1, the inhibitor of MMP-9. Indeed, the ratio of MMP-9/TIMP-1 provides indication about the severity of pulmonary diseases.

5.       Lines 490-492: Considering the similarities between HD exposure and COPD (Chronic Obstructive Pulmonary Disease) pathogens and limited available treatments, it would be of interest if the authors could provide a brief hint on this into the discussion paragraph by introducing slightly COPD (see PMID: 33007849).

6.       Supplementary files were not available neither in the susy platform or at the web link in the manuscript (line 618). Thus, I was not able to evaluate them. Anyway, in my opinion the data reported in the supplementary files have to be mentioned in the “result” section and then argued into the “discussion” one.

MINOR ISSUES

1       The quality of figure 5 need to be improved. It is difficult to read

Author Response

The authors would like to thank the reviewer for their comments. Based off their comments, we have made changes that we believe greatly improve the work. Comments to specific questions are below.

  1. Lines109-110: How did the authors chose the HD concentration values used for their analysis? Do they recapitulate a mean or acute exposure?

To the best of our knowledge, experiments using HD have not been performed with this lung tissue model.  Due to this fact, the initial HD concentrations were selected to provide a wide range of potential exposures (0.01 – 2.5 mg/mL; 0.0629-15.7 mM) that could be handled safely in our laboratory. Existing cell culture data has demonstrated a wide variety of effective concentrations of HD. These findings do not always correlate to the effective doses seen in more complex three-dimensional tissue models containing multiple cell types. In these experiments, we demonstrated the HD induces cellular death in the EpiAirwayTM model in a dose- and time-dependent manner. The concentration of 0.1 mg/mL (629 µM) was selected for the more extensive analysis because this concentration induced approximately 50% cell death 24 hours following the exposure. This would ensure that we would be able to see clear differences between the control and exposed groups for the multi-omic analyses.

  1. Lines 340-342: It would be suitable if authors could corroborate the indication of the apoptosis by means of appropriate assays (e.g., TUNEL)

We have added a supplemental figure from a phase contrast microscope of tissue following HD exposure which show morphological changes associated with apoptosis (i.e. cellular blebbing). Also, the appearance of pyknotic and fragmented nuclei indicate the presence of apoptosis following HD exposure. Much of the literature indicates that HD induces apoptosis and these characteristics have been demonstrated in several different cell culture models (Hayden et al., Toxicol In Vitro. 2009 Oct;23(7):1396-405.doi: 10.1016/j.tiv.2009.07.021). 

  1. Lines 394-405: Why did the authors consider only dysregulation of TFPI, MMP-9 and PLAU among those altered by HD exposure resulting from proteome analysis? Please, motivate the rationale and add it into the manuscript.

We have adjusted the focus of the dysregulated proteins to revolve around the plasminogen system for clarification. For MMP-9, we have added a figure according to your 4th comment to show the ratio between TIMP1 and MMP-9.

  1. Lines 394-405: Given the increase in MMP-9 protein levels, it would be of interest if the authors could provide indications about the expression levels of TIMP1, the inhibitor of MMP-9. Indeed, the ratio of MMP-9/TIMP-1 provides indication about the severity of pulmonary diseases.

We have added a figure that shows the relationship between the MMP-9 and TIMP1 levels and how they track with lung injury. Thank you for your comments as this strengthens the paper.

  1. Lines 490-492: Considering the similarities between HD exposure and COPD (Chronic Obstructive Pulmonary Disease) pathogens and limited available treatments, it would be of interest if the authors could provide a brief hint on this into the discussion paragraph by introducing slightly COPD (see PMID: 33007849).

      The reviewer is correct in recognizing similarities between COPD and HD exposure, and there is a large body of work that reports on the long-term health effects of victims of HD exposure, including developing COPD, asthma, and lung-related cancers (DOI: 10.3109/08958378.2015.1114056). However, it is not within the scope of the work presented here to compare HD exposed tissues and COPD samples. We have added a statement of how uPAR is linked as a biomarker to COPD though.

  1. Supplementary files were not available neither in the susy platform or at the web link in the manuscript (line 618). Thus, I was not able to evaluate them. Anyway, in my opinion the data reported in the supplementary files have to be mentioned in the “result” section and then argued into the “discussion” one.

Thank you for your comments. We do not agree with this assessment of supplemental figures being first introduced in the results section. If the editor has an issue with this decision, they may contact us to discuss.

MINOR ISSUES

1       The quality of figure 5 need to be improved. It is difficult to read.

Figure 5 has been replaced.

Round 2

Reviewer 1 Report

Dear Authors,

I do not have more comments or suggestions and I agree for the pubblication.

Reviewer 3 Report

The authors addressed the issue moved.